# Class-aware Initialization of Early Exits for Pre-training Large Language Models

**Alperen Görmez**[1]   **Erdem Koyuncu**[1]

## Abstract

We propose a novel class-aware weight initialization technique for early exit large language models with the purpose of accelerating pre-training. Our design utilizes the neural collapse phenomenon combined with a Gaussian mixture model for the distribution of feature vectors at a given layer. Specifically, we calculate the average of token representations at the early exit point and use the resulting vectors together with class probabilities for initializing the early exit vectors. The next token prediction accuracy of our class-aware initialization technique is up to five times higher than other baselines at epoch zero and matches or surpasses them in later epochs throughout the pre-training process.

## 1. Introduction

State-of-the-art large language models (LLMs) have a large number of parameters, and generally, the higher the number of parameters, the better the performance (Sutton, 2019; Brown et al., 2020; Zhang et al., 2022; Touvron et al., 2023; Jiang et al., 2024; Gemini Team, 2024; OpenAI, 2024; Meta, 2024). However, their large size and autoregressive design results in high inference latency, which is not desirable for low resource environments and time sensitive settings.

The vast majority of LLMs have a "tunnel-like" architecture: The input to the model is processed by all of the layers in a sequential manner, regardless of the input's inherent difficulty (Kaya et al., 2019; Görmez et al., 2022). On the other hand, not all inputs have the same level of difficulty. Early exit networks exploit this heterogeneous difficulty of inputs. One or more intermediate classifiers are attached to the model, allowing token-level conditional computation (Panda et al., 2016; Teerapittayanon et al., 2016; Kaya et al.,

2019; Görmez et al., 2022; Schuster et al., 2022; Del Corro et al., 2023; Bae et al., 2023; Zhu et al., 2024). Easy tokens can exit early from the LLM in order to save computation.

While the addition of early exits can reduce the inference latency, initially they do not possess the optimal weights. The early exit layers have to be trained first, before being effective at inference time. Early exit layers are typically trained together with the backbone model, with two primary approaches: Training only the early exit and the final exit layers while freezing the non-exit layers, or training the backbone and the exits together (Teerapittayanon et al., 2016; Kaya et al., 2019). Generally, the latter performs better since everything is optimized jointly, but the cross-talk between the exits of the network may lead to suboptimal learning and long training times (Kaya et al., 2019). Ideally, we would like to initialize the weights of the early exit layers in such a way that the cross-talk is minimized, the joint training is facilitated and training time is reduced.

The sizes of both the state-of-the-art LLMs and their training data lead to long training time and high costs. This makes training an early exit LLM even more difficult and costly. In this work, we propose a novel class-aware early exit initialization technique for early exit LLMs to reduce the pre-training costs. We make connections to the optimal detection for the vector additive white Gaussian noise (AWGN) channel from the digital communications domain and utilize the *neural collapse phenomenon* (Papyan et al., 2020). Specifically, we calculate the average of token representations at the early exit point and use the resulting vectors for the initialization. While calculating the average of vector representations has been shown to work well as a decision mechanism for early exit networks (Görmez et al., 2022; Görmez & Koyuncu, 2024), our work is the first to apply it to the initialization of early exit LLMs with the purpose of accelerating pre-training.

We demonstrate the effectiveness of our novel weight initialization technique on WikiText-2 (Merity et al., 2016) and BookCorpus (Zhu et al., 2015) datasets using OPT (Zhang et al., 2022) and TinyLlama (Zhang et al., 2024) models. Notably, our class-aware initialization technique achieves $5\times$ the performance of other baselines at epoch zero. Moreover, it can match or surpass the other baselines at later

[1]Department of Electrical and Computer Engineering, University of Illinois Chicago, Chicago, IL, USA. Correspondence to: Alperen Görmez <agorme2@uic.edu>.

Accepted to the Workshop on Advancing Neural Network Training at International Conference on Machine Learning (WANT@ICML 2024).

epochs throughout the pre-training.

The rest of the paper is organized as follows: In Section 2, we provide a summary of the literature. Then, we establish the notation used throughout the paper in Section 3 and provide a background on the optimal detection problem from the digital communications domain, which we will make connections to later on. In Section 4, we describe our class-aware early exit initialization technique. Finally, we present the results of our experiments in Section 5.

## 2. Related Work

### 2.1. Early Exit LLMs

Numerous attempts have been made in the past to reduce the inference latency of transformer (Vaswani et al., 2017) based models in the past. Perhaps the most visited idea has been adding early exits to BERT variants (Devlin et al., 2018; Zhou et al., 2020; Xin et al., 2020; 2021; Zhu, 2021). However, these models are primarily designed for classification tasks such sentiment analysis, rather than language modeling and text generation.

Developing early exit LLMs for text generation is more challenging, because token-level early exiting requires careful consideration (Elbayad et al., 2019; Liu et al., 2021). Copying hidden states of tokens that exited early to the deeper layers for KV-caching, which confidence measure to use and batch inferencing have been tackled in the past (Schuster et al., 2022; Bae et al., 2023; Del Corro et al., 2023).

### 2.2. Efficient LLM Training

As the number of parameters in an LLM grows, fine-tuning it on datasets becomes more time-consuming and expensive. To address this challenge, researchers have explored parameter-efficient fine-tuning techniques such as adapter approaches, often coupled with quantization (Hu et al., 2021; Liu et al., 2022; Zhang et al., 2023; Dettmers et al., 2024). These methods involve training only a subset of the model parameters, effectively reducing the overall training cost. Most recently, parameter-efficient fine-tuning of early exit LLMs has been explored via data, tensor and pipeline parallelism (Chen et al., 2023; Pan et al., 2024).

## 3. Preliminaries and Problem Formulation

In this section, we establish our notation and lay the foundation for our method by describing the pre-training process of a decoder-only LLM. We then provide background on the problem of optimal detection for the vector AWGN channel from the digital communications domain, which we will make critical connections to later on.

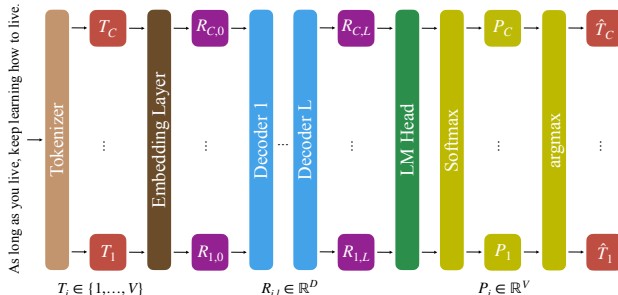

*Figure 1.* Feed-forward phase of pre-training a decoder-only language model.

### 3.1. Pre-training

We focus on the pre-training phase of models belonging to the family of decoder-only LLMs. The model consists of an embedding layer, $L$ decoder blocks and a language modeling (LM) head. Let $V$, $D$, $C$ denote the vocabulary size, the embedding dimensionality, and the context length, respectively.

During the pre-training process, the tokenizer breaks down a text from the training set into $C$ tokens, denoted as $T_i \in \{1, \dots, V\}$, where $i \in \{1, \dots, C\}$. Let $S_v$ denote the set of all training tokens $T_i$ at any position $i$ such that $T_i = v$.

The tokens are subsequently passed through the embedding layer in parallel, resulting in corresponding vectors $R_{i,0} \in \mathbb{R}^D$. These vectors are then fed into the first decoder block, generating output vectors $R_{i,1} \in \mathbb{R}^D$. This iterative process continues sequentially, with the output $R_{i,l}$ of decoder block $l$ being passed to decoder block $l+1$ as the input, where $l$ ranges from 1 to $L-1$.

In the final stage of the feed-forward process, the output $R_{i,L}$ of the last decoder block is fed to the LM head, which is a linear layer with weight matrix $W \in \mathbb{R}^{D \times V}$. The output of the LM head is converted to probability vectors $P_i \in \mathbb{R}^V$ via the softmax operation. Suppose the index of the maximum probability in $P_i$ is $\hat{T}_i$. Since the primary objective of the pre-training phase is next-token prediction, the model is optimized with the cross-entropy loss to ensure $\hat{T}_i = T_{i+1}$. This process is shown in Figure 1.

In order to accelerate inference, one or more early exit LM heads can be integrated to the already pre-trained decoder-only language model. However, the integration of the additional layer(s) necessitates a separate pre-training, which may incur substantial costs as discussed in Section 1. Here, we assume that only one early exit LM head is added. Suppose that this LM head appears after decoder block $K$ where $K < L$, and its weight matrix is $\overline{W} \in \mathbb{R}^{D \times V}$, sharing the same dimensions with the backbone LM head. Our main goal is to find a smart way of initializing $\overline{W}$ such that the

pre-training phase for the early exit LM head will be accelerated, and therefore training costs associated with it will be decreased. Our proposed solution relies on the problem of optimal detection for the vector AWGN channel.

### 3.2. Optimal Detection for the Vector AWGN Channel

The vector AWGN channel can be modeled as

$$r = s_m + n, \quad m \in \{1, \ldots, M\}, \qquad (1)$$

where $r$, $s_m$ and $n$ are $N$-dimensional vectors. A message $s_m$ is sent to the receiver through the AWGN channel, which adds a noise $n$ to the message. The components of the noise vector are independent and identically distributed Gaussian random variables with zero mean and $\frac{N_0}{2}$ variance. The receiver observes $r$, and decides which message was sent among $\{s_1, \ldots, s_M\}$. The goal is to minimize the probability of error. Using the Bayes rule, the optimal detection rule can be written as

$$
\begin{aligned}
\hat{m} &= \arg\max_{1 \le m \le M} \left[ P(s_m \mid r) \right] \\
&= \arg\max_{1 \le m \le M} \left[ \frac{P(s_m) P(r \mid s_m)}{P(r)} \right] \\
&= \arg\max_{1 \le m \le M} \left[ P(s_m) P(r \mid s_m) \right].
\end{aligned}
\qquad (2)
$$

As in Equation 4.2-15 from (Proakis & Salehi, 2008), the equation above can be simplified further as follows:

$$
\begin{aligned}
\hat{m} &= \arg\max_{1 \le m \le M} \left[ P(s_m) P(r \mid s_m) \right] \\
&= \arg\max_{1 \le m \le M} \left[ P(s_m) P_n(r - s_m) \right] \\
&= \arg\max_{1 \le m \le M} \left[ P(s_m) \left( \frac{1}{\sqrt{\pi N_0}} \right)^N e^{-\frac{\|r - s_m\|^2}{N_0}} \right] \\
&= \arg\max_{1 \le m \le M} \left[ P(s_m) e^{-\frac{\|r - s_m\|^2}{N_0}} \right] \\
&= \arg\max_{1 \le m \le M} \left[ \ln P(s_m) - \frac{\|r - s_m\|^2}{N_0} \right] \\
&= \arg\max_{1 \le m \le M} \left[ \frac{N_0}{2} \ln P(s_m) - \frac{1}{2} \|r - s_m\|^2 \right] \\
&= \arg\max_{1 \le m \le M} \left[ \frac{N_0}{2} \ln P(s_m) - \frac{1}{2} \|s_m\|^2 + r \cdot s_m \right] \\
&= \arg\max_{1 \le m \le M} \left[ \eta_m + r \cdot s_m \right],
\end{aligned}
\qquad (3)
$$

where $\eta_m = \frac{N_0}{2} \ln P(s_m) - \frac{1}{2} \|s_m\|^2$. The careful reader will notice the striking similarity between the last line of Equation (3) and the operational logic of a linear layer serving as a classification head. Given an input $x$, the linear layer with weights $w$ and biases $b$ classifies the input according to the maximum element of $b + x \cdot w$.

## 4. Method

Our aim is to initialize the early exit LM head in such a way that it starts from a reasonably good point and achieve a certain level of next-token prediction accuracy before any pre-training, rather than starting from a random point achieving a low next-token prediction accuracy.

We calculate the average of all output vectors after decoder $K$ that correspond to the tokens in $S_v$:

$$M_v = \frac{1}{|S_v|} \sum_{T_i \in S_v} R_{i,K}. \qquad (4)$$

Note that the backbone model is already pre-trained at this point, therefore the intermediate representations are not bad representations. The underlying idea behind Equation (4) is the *neural collapse phenomenon* (Papyan et al., 2020): The intermediate representation of an input belonging to a certain class converges to its corresponding class mean in the final layer of the network. Here, we carry this idea one step further and postulate that, if the input token $T_i$ satisfies $T_i \in S_v$ for some class/word $v$, then the corresponding feature $R_{i,K}$ at layer $K$ is a Gaussian random vector with mean $M_v$ (i.e. the class mean in (4)), and covariance $\frac{N_0}{2} \mathbf{I}$, where $N_0$ is a hyperparameter to be tuned experimentally. Now suppose that the early exit LM head is the receiver we mentioned in Section 3.2. In this context, we can write

$$R_{i,K} = M_v + \epsilon, \quad v \in \{1, \ldots, V\}, \qquad (5)$$

where $R_{i,K}$, $M_v$, and $\epsilon$ are all the $D$-dimensional vector. Also, $\epsilon$, is a zero-mean noise vector with covariance $\frac{N_0}{2} \mathbf{I}$. The mean vector $M_v$ is sent to the early exit LM head as the message, and noise $\epsilon$ has been added during transmission. The early exit LM head observes $R_{i,K}$, and decides which mean vector was sent among $\{M_1, \ldots, M_V\}$.

Similar to Equation (3), the optimal decision equation for the early exit LM head can be written as

$$
\begin{aligned}
\overline{T}_i &= \arg\max_{1 \le v \le V} \left[ \eta_v + R_{i,K} \cdot M_v \right], \ i = 1, \ldots, C \\
\eta_v &= \frac{N_0}{2} \ln P(M_v) - \frac{1}{2} \|M_v\|^2,
\end{aligned}
\qquad (6)
$$

where $N_0$ is a hyper-parameter and $P(M_v)$ is the prior probability for each token in the training set, determined using the empirical frequencies in the training set.

As a result, the early exit LM head is initialized as

$$\overline{W} = [M_1, \ldots, M_V] \in \mathbb{R}^{D \times V}, \qquad (7)$$

with a separate bias vector $\eta = [\eta_1, \ldots, \eta_V]$. Our initialization method is shown in Figure 2.

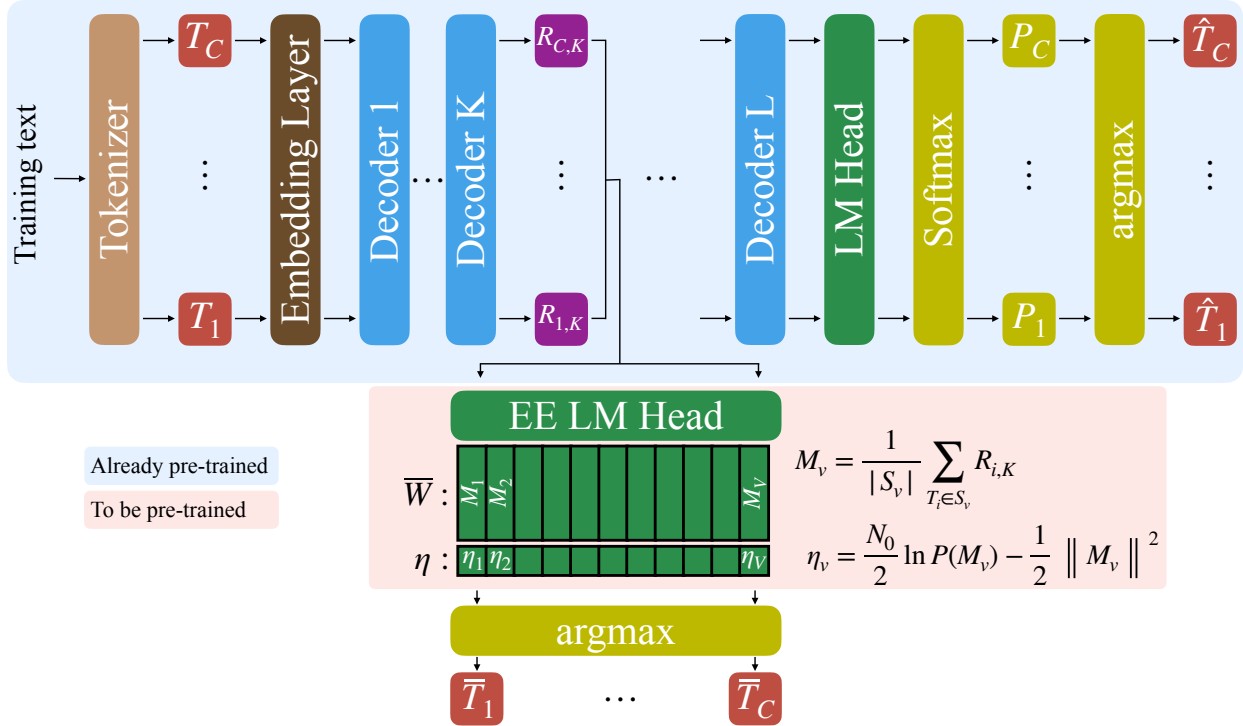

*Figure 2.* Our proposed method for initializing the early exit LM head using the mean representation vectors for each token in the vocabulary set.

*Table 1.* Summary of the models used in our experiments.

| MODEL | $L$ | $D$ | $V$ |
|---|---|---|---|
| OPT-125M | 12 | 768 | 50272 |
| OPT-350M | 24 | 1024 | 50272 |
| OPT-1.3B | 24 | 2048 | 50272 |
| TINYLLAMA-1.1B | 22 | 2048 | 32000 |

## 5. Experiments and Results

In this section we describe our experiments in detail and present the numerical results.

### 5.1. Models

In our experiments, we used OPT-125M, OPT-350M, OPT-1.3B models from the OPT model family (Zhang et al., 2022); as well as TinyLlama-1.1B (Zhang et al., 2024).

The OPT model family is a series of open-sourced decoder-only language models ranging from 125M to 175B parameters. The largest OPT model performs similarly to GPT-3 (Brown et al., 2020) with approximately $1/7^{th}$ of the training cost (Zhang et al., 2022).

The TinyLlama1.1B model is developed with the goal of pre-training such a compact model on 3 trillion tokens (Zhang et al., 2024). The model shares the same architecture with Llama2 model family (Touvron et al., 2023).

The number of decoder layers ($L$), embedding dimensionality ($D$) and vocabulary size ($V$) of the models we used in our experiments are shown in Table 1.

### 5.2. Datasets

In our experiments, we used the WikiText-2 (Merity et al., 2016) and the BookCorpus (Zhu et al., 2015) datasets for pre-training the models.

WikiText-2 is a collection of tokens extracted from the verified "Good" and "Featured" articles from Wikipedia (Merity et al., 2016). For pre-training, we used the "wikitext-2-v1" subset from HuggingFace, which contains 44.8K rows.

BookCorpus dataset is a large collection containing more than 11K books and 74M rows (Zhu et al., 2015). Due to its size, we used 1% of the dataset. We allocated 80% of the 1% for pre-training, and the remaining 20% for evaluation.

### 5.3. Experiment Settings

For a backbone model, we begin by downloading the most recent checkpoint from HuggingFace. This checkpoint is the result of training the model on a large and diverse collection of datasets. However, since we are going to add an early

exit layer and pre-train it on only one dataset, we fine-tune the backbone model on our dataset for 3 epochs so the effect of other datasets on the model is minimal. We found out that fine-tuning for more than 3 epochs led to overfitting.

After the initial fine-tuning of the backbone model, we add the early exit LM head after decoder $K = L/2$. Specifically, $K$ is 6, 12, 12 and 11 for OPT-125M, OPT-350M, OPT-1.3B and TinyLlama-1.1B models respectively. The early exit LM heads share the same architecture and number of parameters with the backbone LM head, the only difference is that we allow the early exit LM head to have a bias vector.

We train the resulting early exit LLM on two different settings. In the first setting, called "no freezing," all parameters are trainable. In the second setting, called "freezing," we freeze the parameters of the model except the two LM heads. These two settings are how early exit neural networks are trained in the literature (Scardapane et al., 2020; Laskaridis et al., 2021). The training is done on a single NVIDIA A6000 with a batch size of 32. Due to limited hardware memory, we used a context length of $C = 128$. We used PyTorch (Paszke et al., 2019) in our experiments.

## 5.4. Results

We now present the results of our class-aware early exit initialization method and compare it against two other initialization techniques:

**1 Random initialization:** This is the default weight initialization technique for linear layers in PyTorch (Paszke et al., 2019). The weights $\overline{W} \in \mathbb{R}^{D \times V}$ are initialized as $\overline{W} \sim U\left[-\frac{1}{\sqrt{D}}, \frac{1}{\sqrt{D}}\right]$, where $U$ is the random uniform distribution.

**2 Copy-from-backbone:** Since the weights $W$ of the backbone LM head is already pre-trained and have the same dimensions as $\overline{W}$, copying $W$ into $\overline{W}$ can serve as a good starting point (Pan et al., 2024).

For our class-aware initialization, we use $N_0 = 0.25$ and we use the empirical frequencies of tokens in the training set for $P(M_v)$, i.e., the number of occurrences of the token divided by the total number of tokens in the training set.

We report the next-token prediction accuracy throughout the pre-training epochs for all initialization techniques. We pre-trained the models for 10 epochs as performance started to drop due to overfitting. The results on the WikiText-2 datasets are shown in Figure 3.

The most important takeaway from Figure 3 is that, our class aware initialization technique achieves 25% next-token prediction accuracy at epoch zero, without any training. On the other hand, random initialization and copying from

backbone can achieve at most 5%. This shows that class-aware initialization of early exits is a promising technique for resource constrained devices and settings.

For the "no freezing" setting, although class-aware initialization starts pretty well, it is surpassed by the copy-from-backbone method easily. There are also some scenarios where random initialization surpasses the class-aware initialization as in Figure 3c and Figure 3e. Here, we can easily match the baselines via a convex combination:

$$\overline{W} = \alpha W_{CA} + (1 - \alpha)W_B, \tag{8}$$

where $W_{CA}$ is the weights initialized in a class-aware manner, and $W_B$ is either random initialized weights or the copied weights from the backbone LM head. This convex combination gets the best of both worlds: It helps preserve the performance of class-aware initialization at epoch zero, and it matches the copy-from-backbone performance at later epochs. In our experiments we evaluated $\alpha \in \{0.2, 0.4, 0.6, 0.8\}$, and we show the best performing $\alpha$-curves in Figure 3.

In the "freezing" setting, only the LM heads are trainable, therefore learning is more difficult. As it can be seen from Figure 3b, Figure 3d, Figure 3f, Figure 3h; the random and copy-from-backbone methods struggle heavily and cannot achieve a good next-token prediction accuracy. On the other hand, our class-aware initialization starts from a pretty good point and keeps performing at the same level throughout the pre-training. Only for the OPT-125M model, there is a sharp drop at the first epoch of the pre-training as seen in Figure 3b. This drop can be somewhat treated by the convex combination equation given in Equation (8).

The same trends are observed for the BookCorpus dataset as seen in Figure 4. Specifically, without any training, the class-aware initialization starts from a high next-token prediction accuracy and the convex combination allows preserving the high performance throughout the pre-training. Notably, in the "freezing" setting, class-aware initialization performs the best

## 6. Conclusion

We developed a novel class-aware weight initialization technique for early exit LLMs based on mean representation of tokens. We made connections to the optimal detection problem for the vector AWGN channel from the digital communications domain. Our method performs better than baselines in both "no freezing" and "freezing" settings. We showed the applicability of our method to various model families and datasets, and its effectiveness on accelerating the pre-training phase.

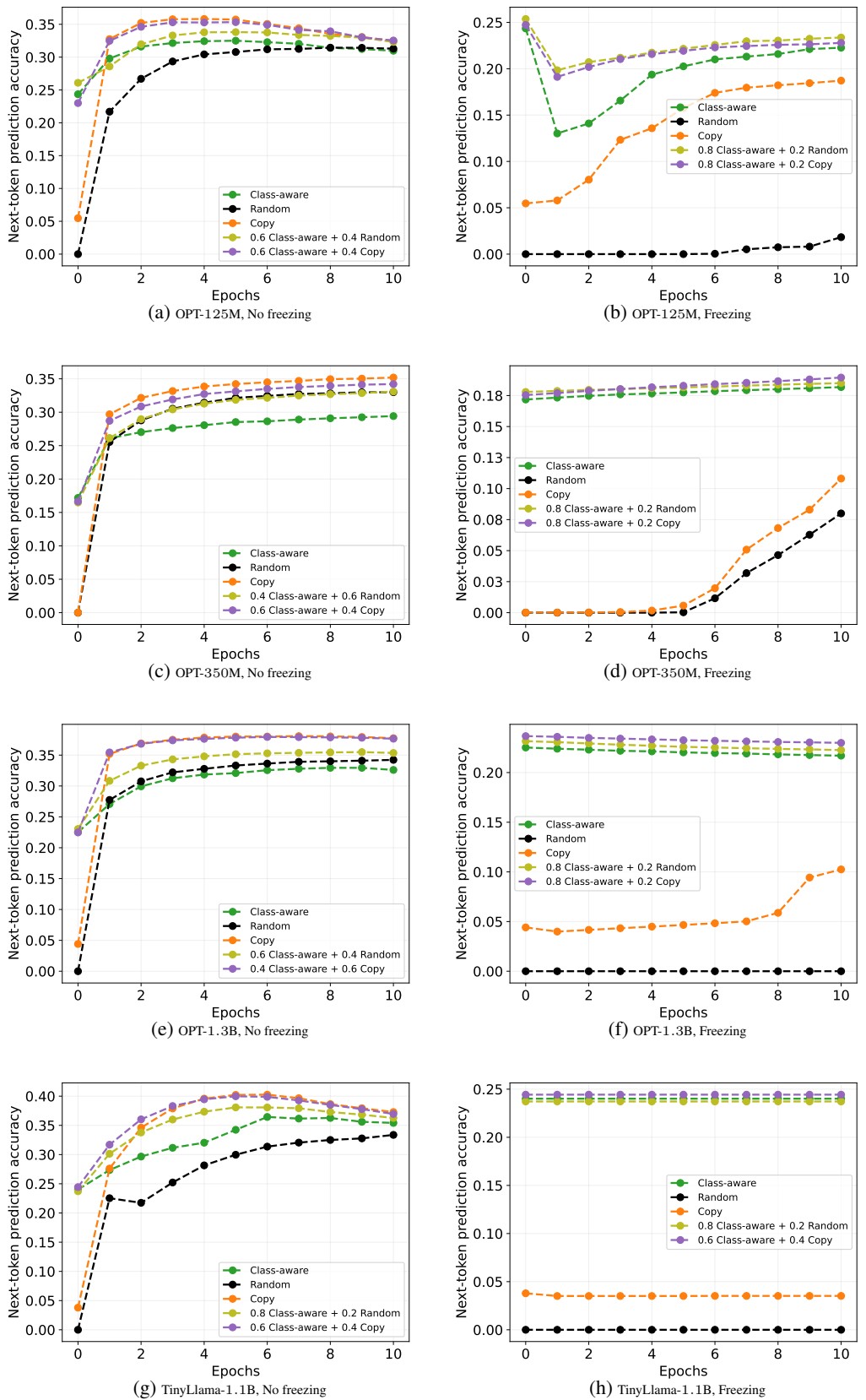

*Figure 3.* Next-token prediction accuracies on WikiText-2 for the early exit LM head initialization techniques.

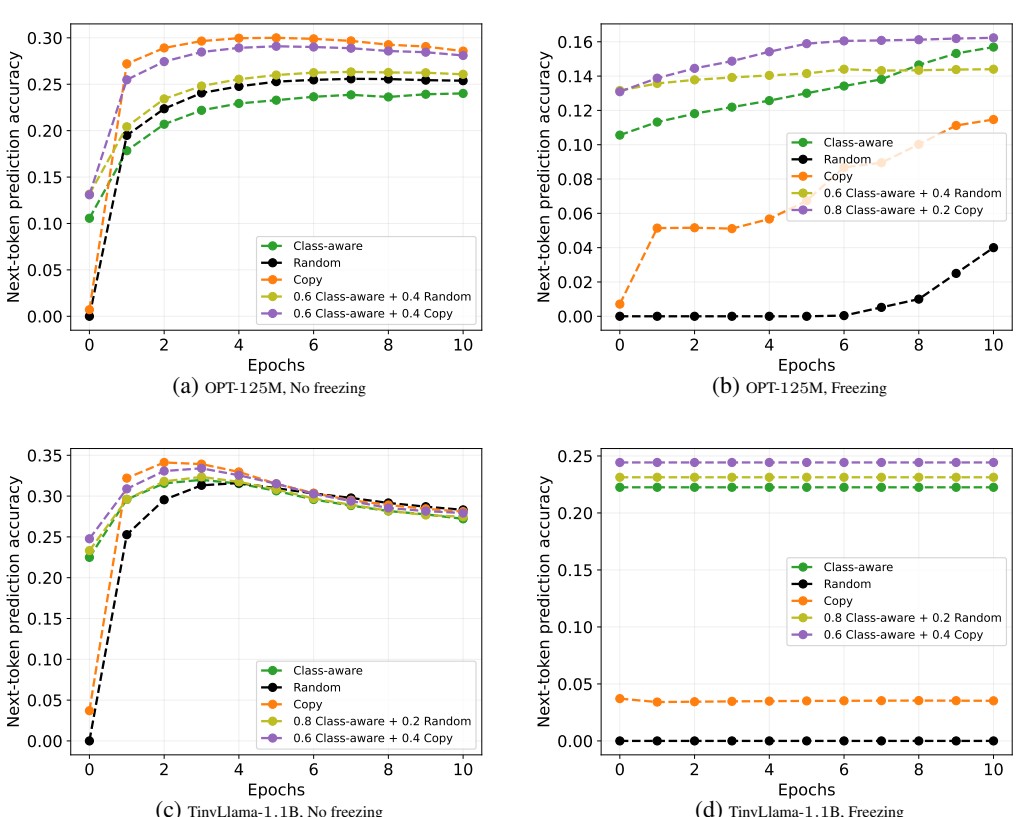

*Figure 4.* Next-token prediction accuracies on BookCorpus for the early exit LM head initialization techniques.

## Acknowledgements

This work was supported in part by the Army Research Lab (ARL) under Grant W911NF-21-2-0272, in part by the Army Research Office (ARO) under Grant W911NF-24-1-0049, and in part by the National Science Foundation (NSF) under Grant CNS-2148182.

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
