# OpenReview forum: "Class-aware Initialization of Early Exits for Pre-training Large Language Models"
_ICML.cc/2024/Workshop/WANT — WANT@ICML 2024 Poster_

### Official Review · Reviewer_w2eU · 2024-06-13

**Confidence:** 4

**Summary:**

This paper proposes an initialization strategy of early exit layers motivated by the Vector AWGN Channel. The method is simple and easy to understand - as it involves computing the weights of a particular token as the average of all output vectors of that token from a reference corpus. Experiments show promising results when there is no training at all. However, I some concerns about evaluation settings and lack of certain analyses.

**Strengths:**

1. The method is simple in terms of implementation.
2. Motivation is easy to understand and follow.
3. It's interesting to se that the method shows reasonable performance out of the box without having to perform any training, but I have some concerns about fairness in comparison with respect to other methods (see weaknesses).

**Weaknesses:**

1. There is no analysis on downstream task performance, the next token prediction accuracy measure will not really capture those aspects, especially how the method impacts the "original" performance of the model compared to others.
2. Since the external corpus plays the main role of initialization in the proposed method, there is a lack of analysis on how the size and quality of the training corpus impacts each method.
3. It's probably not fair to compare the proposed method with the other methods at epoch 0, since it technically sees all the data of the target corpus during initialization.
4. It's interesting that  Copy catches up in a single epoch in the non-frozen setting – suggesting it may only need  to learn some ‘scaled’ features to be as good in the frozen setting. Hence, I wonder why the authors did not try what happens when the copy method is used, but the weights at only Decoder K=L/2 are kept trainable.

**Suggestions:**

1. Why pre-train on multiple epochs of small corpus rather than single epoch but more data? (since the latter is more common practice)
2. The presentation of the paper, such as figure placements, has room for improvement.
3. Period missing in line 262 (before conclusion).

---

### Official Review · Reviewer_JvED · 2024-06-14
**Review of Submission #36**

**Confidence:** 3

**Summary:**

The authors introduce a novel strategy to initialize early exit heads that speeds up pre-training compared to random initialization and classification head initialization. The authors motivate their strategy from neural collapse frameworks. Under k-class data and gaussian noise assumption, they show the optimality of their strategy. Finally, under 4 architectures and 2 pre-training settings, they showcase the utility of their proposed framework.

**Strengths:**

The strength of the paper lies in its motivation and the exposition to their proposed method. The authors build on existing frameworks of neural collapse and show that a mean of token embeddings can prove to be a good initialization for early exit heads. Furthermore, they carefully initialize their biases, building up on their assumption of gaussian noise in the token embeddings. With a careful experimental study, the authors discuss the utility of their proposed framework.

**Weaknesses:**

The major concern of the proposed framework is that it works better than random initialization only when the rest of the model is frozen during pre-training. The authors require convex combination of different initializations for performance gains in the non-freezing case.

For future versions, the authors can provide more explanations/fixes of instability issues in some of the experiments, where the performance drops drastically after few epochs of training. Furthermore, the authors can regularize the model when they observe overfitting with training. Finally, it will be interesting to see the robustness of their method to initializing the early exit head at different layers of the model.

---

### Official Review · Reviewer_8ofm · 2024-06-16
**A Simple method but with some limitations**

**Confidence:** 4

**Summary:**

The paper proposes a technique to develop early exit networks from regular pre-trained base LLMs. Moreover the proposed technique is classified as a initialization technique which the authors claims to solve representation collapse during traininig. Each head is initialized with mean representation of the pre-vious blocks and the authors have provided an intuitive explanation of the efficacy of this technique.

**Strengths:**

The key strengths are listed below.

* The problem of developing a regular pre-trained decoder style LLM into an early exiting LLM seems important for faster generations.

* The paper is written without much typos and grammatical errors and the content flow is coherent.

**Weaknesses:**

Some of the key limitations are listed below.

* The title and the abstract mentions pre-training but the paper uses already pre-trained models as backbones and then trains it for some epochs. Finally they add their initialization technique to make an already well trained LLM ready for early exiting. The setting looks a bit unconvincing. And this paper needs a good re-write to present the motivation and problem setting.

* The datasets used here is very small and no base LLMs can be pre-trained with such datasets.

* The baselines are unclear i.e. the rationale behind why these methods are used for apples apples comparision is unclear.

* A major limitation in my opinion is the motivation- This paper pre-dominantly uses papyan et al. as a motivation for neural collapse. The referred paper used small scale vision models with MNIST type small datasets (which has explicitly defined classes) to study the neural collapse phenomenon. Now, the authors have not established that such a phenomenon exists in LLMs where the classes are not explicitly defined in pre-training data. It would be more convincing if the authors start with first establishing this phenomenon for LLMs that too during pre-training.

* More early exiting baselines can help establishing the efficacy of this work such as- https://arxiv.org/abs/2207.07061.
* More initialization baselines are required.

**Limitations:**

See my comments about the weakness.

**Suggestions:**

See my comments about the weakness.

---

### Meta-Review · Area_Chair_NM9E · 2024-06-18

**Recommendation:** Accept (Poster)
**Confidence:** 4

**Metareview:**

Overall reviewer sentiment for this submission appears to be positive. Some common strengths and weaknesses pointed out by reviewers:

* (+) Multiple reviewers find that the work addresses an important and relevant problem.
* (+) The proposed method is relatively simple and easy to implement.
* (+) Paper is well-written and easy to follow, although there's room for improvement.
* (+) Reasonable experimental evaluation and zero-shot results; multiple minor issues, as listed below.
* (-) Experimental settings: reviewers point out issues with the size and scale of the datasets, lack of evaluation on downstream tasks, and fairness of comparisons (eg: comparing at epoch 0).
* (-) One reviewer points out that the basis for the work (neural collapse) hasn't been verified on larger networks and datasets.
* (-) Baselines: reviewers note the lack of comparisons to appropriate baselines (eg: Schuster et al.)

I recommend acceptance (poster), and request the authors to incorporate the changes suggested by reviewers.

---

### Decision · Program_Chairs · 2024-06-18

**Decision:**

Accept (Poster)

**Comment:**

We thank the authors for their time and contribution to WANT and we are pleased to share that after the reviewing process the paper has been accepted. Congratulations! We encourage the authors to consider reviewers' feedback for the improvement of the camera-ready version. We hope to see you in person at the workshop and brainstorm on efficient training research together!